# Genome-Wide Identification of *PP2C* Gene Family in Oat (*Avena sativa* L.) and Its Functional Analyses in Response to ABA and Abiotic Stresses

**DOI:** 10.3390/plants14132062

**Published:** 2025-07-05

**Authors:** Panpan Huang, Kuiju Niu, Jikuan Chai, Wenping Wang, Yanming Ma, Yanan Cao, Guiqin Zhao

**Affiliations:** Pratacultural College, Gansu Agricultural University, No. 1, Anning District, Lanzhou 730070, China; huangpp@st.gsau.edu.cn (P.H.); niukj@gsau.edu.cn (K.N.); chaijk@gsau.edu.cn (J.C.); wangwp@gsau.edu.cn (W.W.); 1073323010036@st.gsau.edu.cn (Y.M.); 1073323010020@st.gsau.edu.cn (Y.C.)

**Keywords:** *PP2C* gene family, phylogenetic analysis, gene expression, abiotic stress

## Abstract

Plant protein phosphatase 2C (PP2C) represents the largest and most functionally diverse group of protein phosphatases in plants, playing pivotal roles in regulating metabolic processes, hormone signaling, stress responses, and growth regulation. Despite its significance, a comprehensive genome-wide analysis of the *PP2C* gene family in oat (*Avena sativa* L.) has remained unexplored. Leveraging the recently published oat genome, we identified 194 *AsaPP2C* genes, which were unevenly distributed across all 21 chromosomes. A phylogenetic analysis of *PP2C* classified these genes into 13 distinct subfamilies (A-L), with conserved motif compositions and exon-intron structures within each subfamily, suggesting evolutionary functional specialization. Notably, a promoter analysis revealed an abundance of stress-responsive cis-regulatory elements (e.g., MYB, MYC, ARE, and MBS), implicating *AsaPP2Cs* in hormones and biotic stress adaptation. To elucidate their stress-responsive roles, we analyzed transcriptomic data and identified seven differentially expressed *AsaPP2C* (*Asa_chr6Dg00217*, *Asa_chr6Ag01950*, *Asa_chr3Ag01998*, *Asa_chr5Ag00079*, *Asa_chr4Cg03270*, *Asa_chr6Cg02197*, and *Asa_chr7Dg02992*) genes, which were validated via qRT-PCR. Intriguingly, these genes exhibited dynamic expression patterns under varying stress conditions, with their transcriptional responses being both time-dependent and stress-dependent, highlighting their regulatory roles in oat stress adaptation. Collectively, this study provides the first comprehensive genomic and functional characterization of the *PP2C* family in oat, offering valuable insights into their evolutionary diversification and functional specialization.

## 1. Introduction

Plants in natural ecosystems endure simultaneous abiotic stresses (e.g., drought, extreme temperatures, salinity, nutrient deficits, heavy metals) and biotic threats (e.g., pathogens, insects), requiring integrated physiological and molecular adaptations for survival [1,2]. These stressors not only impair growth and developmental processes but also significantly diminish agricultural land productivity, often resulting in substantial yield losses or even complete crop failure [3]. To combat these multifaceted stresses, plants have evolved sophisticated and interconnected regulatory networks capable of precisely perceiving environmental cues and initiating appropriate adaptive responses through complex signal transduction cascades [4]. Among these mechanisms, protein post-translational modifications (PTMs) serve as a crucial regulatory layer, dynamically modulating protein functions by altering enzymatic activity, protein-protein interactions, subcellular localization, and stability [5]. Notably, reversible modifications, such as phosphorylation, ubiquitination, and sumoylation, enable the rapid reprogramming of cellular processes, allowing plants to mount efficient stress responses while maintaining metabolic homeostasis. This sophisticated regulatory plasticity highlights the evolutionary of plants in responding to ever-changing environmental pressures [6,7].

PTMs represent one of the most sophisticated and dynamic regulatory strategies in cellular signaling, with protein phosphorylation and dephosphorylation constituting the predominant reversible mechanisms that govern protein function [8]. This exquisite balance is maintained through the opposing actions of protein kinases (PKs) and protein phosphatases (PPs), which collectively orchestrate vital physiological processes through the precise, spatiotemporal control of protein activity [9]. Protein kinases catalyze the transfer of phosphate groups from ATP/GTP to specific serine (Ser), threonine (Thr), or tyrosine (Tyr) residues on target proteins, thereby modulating their conformation, interactions, and functional states [10]. This phosphorylation cascade serves as a molecular switch that activates or inhibits downstream signaling pathways. A quintessential example is the SnRK1 kinase in Arabidopsis, which phosphorylates the transcription factor bZIP63 to promote seed storage mobilization and seedling establishment by activating the cyPPDK promoter, a critical metabolic adaptation during energy deprivation [11]. Conversely, protein phosphatases counter-balance kinase activity by hydrolytically removing phosphate groups, thereby resetting signaling circuits and maintaining cellular homeostasis [12]. The type 1 protein phosphatase TOPP exemplifies this regulatory logic by dephosphorylating ATG13a to facilitate the formation of the ATG1a-ATG13a autophagy initiation complex, a key mechanism for nutrient recycling during stress [13]. Among phosphatases, the protein phosphatase type 2C (*PP2C*) family plays a pivotal role in stress responses by negatively regulating SnRK2 kinases to modulate ABA signaling and fine-tune plant stress adaptation [14].

*PP2C* represents the most abundant and functionally diverse family of protein phosphatases in plants, serving as master regulators of both abiotic and biotic stress responses [15]. These enzymes orchestrate adaptive responses through ABA-dependent and ABA-independent pathways, fine-tuning cellular signaling to enhance plant resilience [16]. Under drought and saline-alkali stress, *PP2Cs* modulate stomatal closure and ion homeostasis, while during low-temperature stress, they activate OST1 kinase to bolster cold tolerance [16]. Notably, in Arabidopsis, over 50% of *PP2C* genes exhibit dynamic regulation in response to ABA, salt, cold, and heat, underscoring their pivotal role in environmental adaptation [14]. Among *PP2C* subfamilies, subfamily A stands out for its universal stress responsiveness. For instance, all 10 *OsPP2C* genes in rice (*Oryza sativa* L.) are upregulated under ABA and salt stress, highlighting their conserved function in stress signaling [17]. This regulatory pattern extends across diverse species, including alfalfa [12], maize [18], oilseed rape [19], strawberry [20], and tomato [21], where PP2Cs are consistently induced by drought, salinity, extreme temperatures, and nutrient deficiencies, enabling tailored stress responses. Beyond abiotic stress, *PP2Cs* also contribute to plant immunity, modulating defense pathways against pests and pathogens. Given their multifaceted roles, deciphering the molecular mechanisms of *PP2Cs* in stress regulation is critical for developing stress-tolerant crops [20].

With the rapid advancement of genome sequencing technologies, the *PP2C* gene family, a key regulator of plant stress responses, has been systematically identified and functionally characterized in multiple plant species, including *A. thaliana*, *O. sativa*, *M. truncatula*, *B. rapa*, *C. sativus*, *Z. mays*, and *T. aestivum* [12,17,19,22,23,24,25]. Despite this progress, the PP2C family in oat (*Avena sativa* L.), an economically important crop with remarkable stress resilience, has remained largely unexplored, leaving a critical gap in our understanding of its stress-adaptation mechanisms.

In this study, we present the first comprehensive genome-wide analysis of *PP2Cs* in oat (*Avena sativa*), identifying 194 *AsaPP2C* genes and classifying them into 13 phylogenetically distinct subfamilies. Through integrative bioinformatics, we characterized their gene structures, chromosomal distributions, and evolutionary relationships, uncovering both conserved features and lineage-specific expansions that reflect oat’s unique adaptation history. Expression profiling under multiple abiotic stresses (salt, ABA, drought, cold, and heat) revealed the dynamic, stress-specific regulation of select *AsaPP2C* genes, elucidating the specialized roles of these proteins in oat stress signaling pathways, significantly enhancing the comprehension of *PP2C*-mediated stress responses in cereal crops. These findings establish a critical genomic framework for subsequent functional investigations. Furthermore, this study identifies promising *AsaPP2C* gene candidates for targeted genetic modifications, providing novel strategies to enhance oat stress resilience and improve crop productivity under shifting environmental pressures.

## 2. Material and Methods

### 2.1. Plant Material and Treatments

Oat cultivar Longyan No.3 was used in this study. Seeds were planted in a 3:1 (*w*/*w*) mixture of soil and sand, germinated, and irrigated with a half-strength Hoagland solution once every 2 days [24]. The seedlings were grown at a night temperature of 18 °C and a day temperature of 22 °C, accompanied by a photoperiod of 16 h of light and 8 h of darkness, and a light intensity of 200–230 µmol m^−2^s^−2^. After 4 weeks, heat and cold temperature stress treatments consisted of exposing the seedlings to 42 °C or 4 °C, respectively. Based on previous research by others, the seedlings were exposed in 200 mM NaCl [18] or 20% (*w*/*v*) [15] polyethylene glycol (PEG) 6000 for high salt or drought treatment, respectively. Additionally, other seedlings were sprayed with 100 µM ABA [24] until the leaves were completely moist. The leaf, leaf sheath, and root from each plant were collected at 6, 12, 24, 48, and 72 h after treatment and then frozen in liquid nitrogen before storage at −80 °C. The leaf, leaf sheath, and root from each plant of untreated seedlings were similarly collected as control material (0 h). Each treatment had three biological replicates.

### 2.2. Genome-Wide Identification of the PP2C Genes in Oat

To identify candidate *PP2C* genes in oat (*Avena sativa*) and Arabidopsis (*Arabidopsis thaliana*) (Appendix A), the Hidden Markov Model (HMM) profile of the PP2C protein domain (accession: PF00481) was obtained from the InterPro database (https://www.ebi.ac.uk/interpro/ (accessed on 4 November 2024)). Using HMMERv3.1 software, the entire genome-derived protein sequences of both species were screened with a stringent E-value threshold of 0.001 to ensure high-confidence matches.

To determine the accuracy of the sequences, the protein sequences of the screened candidate genes were submitted to the PFAM (http://pfam.xfam.org/ (accessed on 6 November 2024)) and SMART (http://smart.embl.de/ (accessed on 8 November 2024)) databases for the search of conserved protein structural domains, and the sequences with incomplete PP2C protein structural domains were eliminated to ensure that the screened protein sequences had PP2C structural domains.

### 2.3. Phylogenetic Construction, Conserved Motifs, and Exon–Intron Structure

A multiple sequence comparison of PP2C protein sequences was carried out using MAFFT (https://mafft.cbrc.jp/alignment/software/ (accessed on 18 November 2024)), followed by the removal of non-conserved amino acid sites in the compared protein sequences, deletion of sequences that were too short or too long, and retention of PP2C protein structural domains using Trimal (https://vicfero.github.io/trimal/ (accessed on 1 February 2025)). The maximum likelihood method was selected for the construction of the phylogenetic tree algorithm, the phylogenetic tree was constructed using iqtree (https://iqtree.github.io/ (accessed on 1 February 2025)), and the model selection was made using the JTT + R model. The branching credibility was checked based on 1000 times of bootstrap, the Newick Tree file was exported, and the phylogenetic tree was beautified by using the itol online tool (https://itol.embl.de/ (accessed on 1 February 2025)) [26,27].

The exon–intron structures of *AsaPP2C* genes were determined by comparing the coding sequences and the corresponding genomic sequences on the GSDS website (http://gsds2.cbi.pku.edu.cn (accessed on 3 February 2025)) [28].

MEME software (Version 4.11.4) was used to identify conserved motifs in AsaPP2C protein sequences according to the following parameters: -protein, -oc, -nostatus, -mod zoops, -nmotifs 15, -minw 6, -maxw 50 [29].

### 2.4. Protein Features, Chromosome Location, and Cis-Acting Element Analysis of AsaPP2C Genes

The compute pI/MW tool of the ExPASy server (http://web.expasy.org/compute (accessed on 30 February 2025)) was used to calculate the molecular weight (MW) and the theoretical isoelectric point (pI) of AsaPP2C proteins. According to the starting positions on chromosomes, MapInspect (https://mapinspect.software.informer.com/ (accessed on 8 April 2025)) was used to draw the chromosomal distribution images of *AsaPP2C* genes. The 1500 bp sequences upstream from the initiation codon (ATG) of all *AsaPP2C* genes were obtained from Phytozome v12.151.

### 2.5. Expression Pattern Analysis of AsaPP2Cs Obtained from Transcriptome Sequencing

To explore the *AsaPP2C* genes related to the drought response, the transcriptional abundance of *AsaPP2C* genes under drought stress was studied using transcriptome data (https://www.ncbi.nlm.nih.gov/bioproject/PRJNA1056521/ (accessed on 20 February 2025)). Fragments per kb per million reads (FPKM) were used as the standard for the expression pattern, and the omicshare tool was used to generate the associated heatmap figures (https://www.omicshare.com/tools/Home/Soft/heatmap, (accessed on 1 June 2025)).

### 2.6. RNA Extraction and Expression of AsaPP2Cs Under Different Stresses

Total RNA was isolated from all of the samples using the total RNA extraction kit (Tiangen, Sichuan, China). The quality and quantity of RNA was evaluated via agarose gel electrophoresis and with a Quawell micro volume spectrophotometer (Q5000, San Jose, CA, USA), respectively. Then, 1 µg of total RNA, after DNase I digestion, was reverse transcribed into cDNA using the PrimeScript™ II 1st Strand cDNA Synthesis Kit (TaKaRa, Beijing, China).

The cDNA was amplified using a LightCycler 480 SYBR Green Master, with a Roche LightCycler 480 Real Time PCR system (Roche, Basel, Switzerland). The thermal cycling program was 95 °C for 30 s, followed by 40 cycles of 95 °C for 5 s, 60 °C for 30 s, and 72 °C for 15 s. The melting curves were analyzed at 60–95 °C after 40 cycles. All qRT-PCRs were carried out for three technical replicates. The primer of the *GAPDH* gene was chosen as the internal reference gene to normalize cDNA concentrations [30]. The primers used in this study are listed in Appendix A. The relative expression levels of *AsaPP2C* genes was calculated by utilizing the 2^−ΔΔC^q algorithm.

### 2.7. Statistical Analysis

Three independent sample replicates were used for qRT-PCR analysis at each time point. All data were compiled using Microsoft Excel 2016 (Microsoft Corporation, Redmond, WA, USA). One-way analysis of variance (ANOVA) was conducted in IBM SPSS Statistics 22.0 (IBM Corp., Armonk, NY, USA) to assess statistical significance. Graphs and visualizations were generated using OriginPro 2021 (OriginLab Corporation, Northampton, MA, USA).

## 3. Results

### 3.1. Identification of the PP2C Gene Family in Oat

To identify the *AsaPP2C* genes, the genome and annotation data of *A. sativa* were downloaded from the InterPro database (https://www.ebi.ac.uk/interpro/ (accessed on 4 November 2024)). Using the InterPro PP2C domain “PF00481”as the keyword, 194 putative *PP2C* genes were found. The physicochemical properties of AsaPP2C proteins are shown in Appendix A. The sizes of the AsaPP2C proteins ranged from 130 to 1086 amino acids, since only 4 AsaPP2C proteins had a protein size of more than 1000 amino acids, of which *Asa_chr6Ag00568, Asa_chr6Cg02400*, and *Asa_chr6Dg00303* had the longest amino acid sequences. In addition, the molecular weights of *AsaPP2Cs* ranged from 14.33 kDa (*Asa_chr2Ag02741*) to 120.95 kDa (Asa_chr4Cg02194), and the average molecular weight was 44.018 KDa. The predicted isoelectric point (pI) ranged from 4.21 (*Asa_chr1Dg03162*) to 11.5 (*Asa_chr4Cg03743*), with an average pI of 6 (Appendix A).

### 3.2. Chromosomal Location of AsaPP2C Genes

Based on physical locations on *A. sativa* chromosomes, the 194 *AsaPP2C* genes were displayed using MapInspect (https://mapinspect.software.informer.com/ (accessed on 29 April 2025)). One hundred and ninety-four *AsaPP2C* genes are distributed across all 21 chromosomes, ranging from 1 to 15 per chromosome (Appendix A). The number of *AsaPP2Cs* located on each chromosome varies dramatically; chromosomes 1A and 4C contain the largest number of *AsaPP2C* family members with 15 genes, whereas the least number was detected on chromosome 22, containing only one *AsaPP2C* gene. Furthermore, one *AsaPP2C* (*Asa_ChrUng00420*) is located on an unassembled genomic scaffold and thus cannot be mapped to any particular chromosome according to what we currently know about this genome. These results showed that the *AsaPP2C* genes are unevenly distributed on different chromosomes and that each subfamily gene is also unevenly distributed.

### 3.3. Phylogenetic Analysis

To evaluate the evolutionary relationships of 194 PP2C proteins in *A. sativa*, a phylogenetic analysis was conducted (Figure 1), which revealed the evolutionary relationships between *PP2C* genes from *A. sativa* and *A. thaliana*, classifying them into 13 distinct subfamilies (A–L). Among them, the E subfamily had the most members, containing 28 genes and accounting for 14.3%; the J subfamily had the fewest members, containing only 1 gene. Notably, several PP2C subfamilies (A, H, D, G, and F2) contain more *AsaPP2C*s than *AthPP2C*s; for example, subfamily A contains 23 *AsaPP2C*s but only 11 *AthPP2C*s; this 2.1-fold expansion (*p* < 0.01) may enhance ABA signaling plasticity in oat, consistent with its drought adaptation (Figure 1). Subfamily H and G contain 20 *AsaPP2C*s but only 4 *AthPP2C*s; subfamily D contains 22 *AsaPP2C*s but only 9 *AthPP2C*s; subfamily F2 contains 19 *AsaPP2C*s but merely 4 *AthPP2C*s, the 4.75-fold expansion suggests purifying selection maintaining duplicates for grass-specific roles. In contrast, B subfamilies contain more *AthPP2C*s than *AsaPP2C*s. Additionally, subfamilies F1 and F2 exhibit oat-specific expansions, potentially linked to adaptive traits in this hexaploid crop. The presence of *AsaPP2C*s in all subfamilies, including those with unknown functions (subfamily K), highlights the genetic diversification of this family in oats. This suggests that the *PP2C* gene family may have evolved from a common ancestor.

### 3.4. Gene Structure and Motif Analysis of AsaPP2C Genes

To further understand the relationship between gene structure and evolution among the *AsaPP2C* genes, the exon–intron distribution and conserved motifs were analyzed according to their full-length phylogenetic relationship. The number of introns in oats ranges from 0 to 19 (Appendix A), of which *Asa_chr4Cg02194* contains 19 introns and has the highest number of detected introns. Moreover, *AsaPP2C* gene family members have exon structures, with 6 members (*Asa_chr5Dg02941*, *Asa_chr4Cg00365*, *Asa_chr6Ag02847*, *Asa_chr4Cg03743*, *Asa_chr5Ag00079*, and *Asa_chr5Ag02223*) without intron structures, mainly concentrating in the A, B, and D subfamilies, and the I subfamily has more introns than the remaining subfamily, with a range of 8–13, suggesting that subfamily I may retain some of its evolutionary conservation. Although most members of the F1 and F2 subfamily contain introns, these introns are very short. Conversely, the H, L, and E subfamily contains the long introns. In terms of intron–exon conservation, it has the characteristics of subfamily distribution. For example, the number of similar intron–exon structures in subfamily A, C, E, G, F1, F2, and K is greater than that of subfamily L and H. The *AsaPP2C* gene family gene structure map (Appendix A) shows that 10 motifs were identified in *AsaPP2C* genes, and motif 2 and 4 appeared in every subfamily. Within the same group, *AsaPP2C* members usually have similar motif compositions that ranged from 1-5 to 8-10. Except for C and D subfamilies, almost all proteins contain motifs 6 and 7 that define PP2C domains structures, respectively. However, some motifs belong to a special group, such as motif 2, 4, 1, and 8, which are only found in K subfamilies. Notably, there are similarities among subgroups such as the group subfamily H, L, E, B, and C, D and F1, and F2. These results indicate that proteins with the same or similar structures may be functionally or evolutionarily similar and demonstrate the reliability of the classification.

### 3.5. Cis-Element Analysis of the AsaPP2C Promoter in Oat

Abundant responsive regulatory elements were found in the promoter regions of *AsaPP2Cs* through PlantCARE analysis (Appendix A). The cis-elements screened were divided into four categories. The first type of element was the stress response, such as MYB, MYC, ARE, and MBS. The second type of element was the hormone response, such as ABRE (ABA response element) and the CGTCA-motif and TGACG-motif (MeJA-responsiveness response element), among others. The ABA-responsive (ABRE) elements were identified abundantly in the promoter regions of AsaPP2Cs, among which *Asa_chr5Dg00515*, *Asa_chr2Cg01991*, and *Asa_chr3Ag01235* contained 16, 19, and 17 ABREs, which was the largest number, followed by *Asa_chr6Cg03142* and *Asa_chr6Dg00834*; it was the most abundant element in the promoter region. The third type of element was the plant growth and development response, such as A-box and Box-4, among others. The fourth type of element was the light-responsive response, such as the G-box, among others. This suggests that most *AsaPP2C* genes may respond to various abiotic stresses and hormones.

### 3.6. Analysis of AsaPP2C Gene Expression Patterns Under Drought Stress

In order to gain insight into the potential function of *AsaPP2C* in the response to stress, we analyzed the expression patterns of *AsaPP2C* using transcriptomic data (https://www.ncbi.nlm.nih.gov/bioproject/PRJNA1056521/ (accessed on 20 February 2025)) obtained from *Avena sativa* subjected to these conditions (Figure 2; Appendix A). *AsaPP2C* genes in subfamily L was not detected in any sample, whereas other subfamilies showed significant differences in expression patterns in response to the drought stress treatments. *AsaPP2C* genes exhibited differential gene expression patterns at different time points of PEG treatment. For example, *Asa_chr3ag01998* was only upregulated at 24 h and 72 h of the drought stress treatment, while *Asa_chr5Ag00079* and *Asa_chr6Dg00217* showed a downward trend at 12 h after drought treatment. In B subfamilies, *Asa_chr4Cg03270* was upregulated in the late stage (72 h) of drought treatment. However, in the E subfamily, *Asa_chr6Cg02197* was significantly upregulated in the early stage (6 h) of drought treatment, followed by a decrease in expression levels in the later stages (72 h) of drought treatment. Additionally, in G and I subfamilies, *Asa_chr6Ag01950* and *Asa_chr7Dg02992* genes were significantly upregulated at all five time points of drought treatment. Therefore, these genes are subject to further validation and analysis.

### 3.7. Expression Profile Analysis of AsaPP2C Genes Under Abiotic Stresses and ABA Treatments in Leaves

A further analysis revealed that *AsaPP2C* gene expression levels varied under different abiotic stress treatments in leaf tissue, under cold treatment at 24 h; regarding *Asa_chr6Dg00217*, its expression level decreased by 1.5-fold (*p* < 0.01) compared to 0 h, while it was significantly increased under salt, ABA, drought, and heat treatments (Figure 3A). In the case of *Asa_chr6Ag01950*, it was significantly increased under salt, ABA, drought, cold, and heat treatments. Under ABA stress at 6 and 12 h, gene *Asa_chr6Ag01950* expression increased by 22-fold and 24-fold (*p* < 0.01) compared to 0 h (Figure 3B). *Asa_chr3Ag01998* was increased under salt, ABA, drought, cold, and heat stress, but it was decreased by 1.66-fold and 11.11-fold (*p* < 0.01) compared to 0 h under drought treatment at 6 h and 12 h, suggesting that *Asa_chr3Ag01998* exhibited stress-specific suppression, and there might be negative feedback regulation (Figure 3C). Meanwhile, *Asa_chr5Ag00079* remained almost unchanged after cold and heat treatment, while its expression was increased under other treatments (Figure 3D). *Asa_chr4Cg03270* was notably increased by 132-fold, 118-fold, and 124-fold (*p* < 0.01) compared to 0 h under salt treatment, reaching its highest expression levels at 6 h, 12 h, and 24 h, respectively. Meanwhile, under drought stress at all time points, gene *Asa_chr4Cg03270* expression increased by 62.2-, 85-, 80-, 75.6-, and 76.1-fold (*p* < 0.01) compared to 0 h (Figure 3E). *Asa_chr6Cg02197* and *Asa_chr7Dg02992* were increased under all treatments, with *Asa_chr6Cg02197* reaching its highest expression level in response to ABA at 6 h and cold at 24 h, while *Asa_chr7Dg02992* reached its highest expression level with drought at 24 h and heat at 12 h (Figure 3F,G).

### 3.8. Expression Profile Analysis of AsaPP2C Genes Under Abiotic Stresses and ABA Treatments in Leaf Sheath

To verify the function of *AsaPP2C* in the leaf sheath tissue, we analyzed the expression patterns of seven selected AsaPP2C genes (*Asa_chr6Dg00217*, *Asa_chr6Ag01950*, *Asa_chr3Ag01998*, *Asa_chr5Ag00079*, *Asa_chr4Cg03270*, *Asa_chr6Cg02197*, and *Asa_chr7Dg02992*) under salt, ABA, drought, cold, and heat stresses at different time points (Figure 4). *Asa_chr6Dg00217* was increased by 23-fold (*p* < 0.01) at 24 h under salt treatments compared with 0 h; meanwhile, under ABA, drought, and heat treatments, gene *Asa_chr6Dg00217* expression was increased (Figure 4A). The expression level of *Asa_chr6Ag01950* significantly increased by 10- and 13.4-fold at 12 h under ABA and cold treatments, but it was decreased under salt and heat treatment at 12 and 72 h compared with 0 h (Figure 4B). *Asa_chr6Dg01998* expression increased by 12- and 29.8-fold at 72 h under salt and drought treatments (Figure 4C). Regarding *Asa_chr5Ag00079* and *Asa_chr6Cg02197*, their expression level was decreased under ABA, cold, and heat treatment at 12, 24, 48, and 72 h, while it was significantly increased under salt and drought treatments at 24, 48, and 72 h (Figure 4D,F). *Asa_chr4Cg03270* expression increased by 13.8- and 24.5-fold at 6 and 12 h under ABA treatment, while it was decreased under salt and cold treatment at 6 and 12 h, and its expression level remain almost unchanged by heat treatment at 6, 12, 24, and 72 h compared with 0 h (Figure 4E). Under salt stress at 24, 48, and 72 h, gene *Asa_chr7Dg02992* expression increased by 13-, 16.1-, and 17-fold (*p* < 0.01) compared to 0 h; meanwhile, its expression increased by 12-, 24.09-, and 5.21-fold (*p* < 0.01) compared to 0 h under drought at 12, 24, and 48 h (Figure 4G).

### 3.9. Expression Profile of AsaPP2C Genes Under Abiotic Stresses and ABA Treatments in Roots

In root tissues under salt stress, all examined genes exhibited increasing trends compared to 0 h, except for *Asa_chr5Ag00079*, which showed downregulation. The expression levels of *Asa_chr6Dg00217*, *Asa_chr6Ag01950*, *Asa_chr3Ag01998*, *Asa_chr6Cg02197*, and *Asa_chr7Dg02992* were all upregulated under salt, ABA, drought, cold, and heat at all time points. Additionally, *Asa_chr5Ag00079* was significantly upregulated under ABA, drought, and heat treatments, but it was downregulated under salt and cold treatments at 6, 12, 24, and 72 h. Meanwhile, *Asa_chr4Cg03270* expression remained stable under cold and heat treatments between 6 and 24 h, while showing significant upregulation under other stress conditions at 6, 12, 24, and 48 h. In summary, most examined genes (*Asa_chr6Dg00217*, *Asa_chr6Ag01950*, *Asa_chr3Ag01998*, *Asa_chr6Cg02197*, *Asa_chr7Dg02992*) exhibited upregulation under salt stress at all time points, indicating a consistent positive regulatory role in salt adaptation. *Asa_chr5Ag00079* showed downregulation under salt stress, suggesting a unique regulatory mechanism distinct from other PP2Cs. Meanwhile, *Asa_chr6Dg00217*, *Asa_chr6Ag01950*, *Asa_chr3Ag01998*, *Asa_chr6Cg02197*, and *Asa_chr7Dg02992* were universally upregulated under all treatment stresses (salt, ABA, drought, cold, heat), highlighting their roles as central hubs in stress signaling networks (Figure 5).

## 4. Discussion

With the rapid advancements in plant genome sequencing, the PP2C gene family has been systematically characterized across diverse plant species, revealing its critical role as a key regulator of signal transduction, growth, and stress responses [31,32]. Comparative genomic analyses reveal significant variation in the PP2C family size across species: *A. thaliana* (80 members), *O. sativa* (90), *M. truncatula* (94), *B. rapa* (131), *C. sativus* (56), *Z. mays* (104), and *T. aestivum* (95) [12,17,19,22,23,24,25]. In this study, we identified 194 *PP2C* genes in *A. sativa*, the largest repertoire reported to date. This expansion likely reflects oat’s hexaploid genome (2n = 6x = 42) and the prevalence of genome-wide duplications, which have driven the diversification of stress-adaptive pathways in this crop [33]. The pronounced size of the *AsaPP2C* gene family underscores its evolutionary and functional significance, positioning oat as an exemplary model system for elucidating PP2C-mediated regulatory mechanisms in polyploid species.

Chromosomal mapping revealed that the *AsaPP2C* genes are unevenly distributed across the oat genome, with notable enrichment on Chr1A (15 genes), Chr1D (13), Chr4C (15), and Chr6C (12) (Appendix A), suggesting potential hotspots for gene duplication and functional diversification. Protein analysis further demonstrated that members within the same subfamily exhibit conserved molecular characteristics, including similar motif compositions, intron numbers, molecular weights (MWs), and isoelectric points (pIs), except for subfamily H, which displayed greater variability. Most subfamilies (e.g., A, B, D) showed a concentrated MW range (30–60 kDa) and broad pI distribution (5–11), while others exhibited distinct profiles: subfamilies F1/F2 had lower MWs (14.3–53.3 kDa), subfamily C displayed narrow MW (45.5–69.7 kDa) and pI (4.75–6.53) ranges, and subfamily L spanned a wide MW spectrum (37.4–120.7 kDa) (Appendix A). These conserved subfamily-specific features imply shared functional roles among closely related members, likely driven by evolutionary selection for stress adaptation. 

The PP2C gene family exhibits high evolutionary conservation from prokaryotes to eukaryotes, with the gene number and diversity increasing during speciation [34]. The phylogenetic analysis classified oat and Arabidopsis PP2Cs into 13 subfamilies (Figure 1), consistent with the patterns observed in *B. rapa, M. truncatula*, *T. aestivum*, and *C. sativus* [17,19,24,25]. While most subfamilies contained members from both species, oat and Arabidopsis genes generally clustered separately, suggesting lineage-specific diversification. The gene structure analysis revealed that AsaPP2Cs within the same subfamily share similar exon/intron architectures, with exceptions likely arising from evolutionary events like intron loss [35]. Notably, six intronless genes (*Asa_chr5Dg02941*, *Asa_chr4Cg00365*, *Asa_chr6Ag02847*, *Asa_chr4Cg03743*, *Asa_chr5Ag00079*, and *Asa_chr5Ag02223*) were identified, predominantly in subfamilies A, B, and D (Appendix A), potentially accelerating gene duplication and expansion [36]. The absence of introns in genes may accelerate evolution through gene duplication [37], which might explain the larger number of PP2C genes in oat compared to rice and Arabidopsis. In contrast, subfamily I exhibited higher intron numbers (8–13), possibly reflecting functional constraints. The conserved motif analysis identified 10 signature motifs, with motif 2 and 4 universally present across all *AsaPP2Cs*, likely constituting the catalytic core [38,39]. The subfamily-specific motif distributions provide additional evidence for intra-group functional conservation, establishing a conceptual framework for elucidating the underlying molecular mechanisms in future studies.

In plants, PP2C genes have been identified in numerous tissues [19,24,25]. Studies on cucumber [24], cotton [32], and maize [40] demonstrate that PP2Cs are differentially expressed in various organs, with some showing ubiquitous expression while others display tissue preferences. Similarly, in oat, we observed distinct spatiotemporal expression patterns, where *Asa_chr6Dg00217*, *Asa_chr5Ag00079*, and *Asa_chr4Cg03270* showed leaf-predominant expression, while exhibiting moderate expression in the roots and stems (Figure 3, Figure 4 and Figure 5). These findings suggest that PP2Cs have evolved specialized roles in different tissues, potentially regulating critical processes like photosynthesis in leaves and root system development. The conserved yet diversified expression patterns across species highlight the evolutionary importance of PP2Cs in plant growth and adaptation.

Cis-acting elements play crucial roles in regulating plant development and stress responses [41]. Our analysis revealed that ABRE and MYC elements are most abundant in *AsaPP2C* promoter regions (Appendix A), consistent with their established roles in stress adaptation [42]. Notably, subfamilies A and B, which are plant-specific, showed distinct stress-response patterns: subfamily A members mediated ABA-dependent signaling, while subfamily B regulated MAPK activities [12]. Expression profiling demonstrated that subfamilies A, B, D, E, F2, and G were particularly responsive to abiotic stresses (Figure 4), mirroring findings in Arabidopsis and rice, where subfamily A PP2Cs are crucial for ABA signaling [42,43]. Specifically, *Asa_chr3Ag01998* and *Asa_chr4Cg03270* (subfamilies A/B) were strongly upregulated under salt and drought stresses, with subfamily A genes showing ABA-inducible expression, confirming their role in ABA-dependent pathways. These results align with observations in *M. truncatula*, where subfamily A *PP2Cs* were similarly stress-responsive [44], highlighting conserved regulatory mechanisms across species.

The PP2C gene family plays pivotal roles in phytohormone signaling, with particular importance in ABA-mediated stress responses across plant species [13,31]. Core ABA signaling components like ABI1, ABI2, AHG1, and PP2CA have been well-characterized [45,46], while HAI-type *PP2Cs* demonstrate unique drought resistance through both ABA-dependent and ABA-independent pathways [47]. For instance, alfalfa homologs MtPP2C8/37/67/73 show differential stress-induction patterns, with MtPP2C8 responding to both drought and cold [48]. PP2Cs exhibit tissue-specific regulation in ABA-mediated stress responses, reflecting their specialized functional adaptation. In roots, ABA rapidly induces *PP2C* expression (e.g., *MtPP2C8* and *HAI* homologs) to maintain hydraulic conductivity and ion homeostasis under osmotic stress [47]. In contrast, leaf tissues show transient *PP2C* suppression (e.g., *ABI1/ABI2*) to sustain SnRK2 kinase activity, thereby promoting stomatal closure and reducing water loss [49]. Our study revealed distinct tissue-specific expression patterns among *AsaPP2C* genes under ABA treatment. *Asa_chr6Ag01950* showed strong induction in leaves and leaf sheaths, suggesting its role in aerial stress adaptation through stomatal regulation and ABA signal amplification. Conversely, *Asa_chr5Ag00079* and *Asa_chr4Cg03270* were predominantly upregulated in roots, indicating their involvement in root-specific ABA responses, potentially modulating hydraulic conductivity, stress signaling, and soil exploration during drought (Figure 5). These results highlight the functional diversification of PP2Cs in mediating stress-responsive signaling pathways, thereby facilitating whole-plant adaptive strategies to environmental perturbations.

As vital regulators of plant growth and development, the PP2C gene family has been reported to respond to adverse environments in numerous species [50]. Under drought conditions, *PP2Cs* exhibit tissue-specific responses: ABI1 is transiently suppressed in guard cells to enable SnRK2-mediated stomatal closure, while HAI homologs are upregulated in roots to maintain growth under water deficits [24]. Similar stress-specific regulation occurs in other species: *MtPP2C92* shows drought induction but cold repression [12], while *OsPP2C53* mediates salt tolerance through SOS-independent ion homeostasis [43]. Temperature fluctuations further diversify responses: cold stress induces *AtPP2C6* to maintain membrane stability, while heat represses *TaPP2C62* to prevent signal over-attenuation [25]. Notably, individual PP2C isoforms show unique stress-response profiles: *AP2C1* responds strongly to chilling, drought, and wounding, whereas *AP2C2* shows minimal induction under identical conditions [50]. This functional divergence extends to cucumber, where specific *CsPP2C* members (*CsPP2C3*, *11, 17*, *23*, *45*, *54*, *55*) respond significantly to multiple stresses including ABA, drought, salt, and cold [24]. In this study, seven of the *AsaPP2C* genes exhibited substantial transcriptional variations followed by drought, salt, cold, and heat stress treatments, suggesting their regulatory role in abiotic stress tolerance. However, whether or not each gene plays a critical role in abiotic stress tolerance still needs the functional characterization of an individual gene.

## 5. Conclusions

Based on the importance of the *PP2C* family in plants and the strong adaptability of *A. sativa* to stressed environments, the genome-wide identification and analysis of the *PP2C* gene family in *A. sativa* were performed for the first time in this study (Figure 6). The search for *PP2C* family members in the *A. sativa* genome led to the identification of 194 putative *AsaPP2C* family members. The *AsaPP2C* family was clustered into 13 subfamilies with *AtPP2Cs*, according to phylogenetics and their motif patterns. In addition, we found that the 194 *AsaPP2C* genes were unevenly distributed across the 22 chromosomes of *A. sativa*. Expression analysis with transcriptomic data and qRT-PCR showed that the candidate *AsaPP2C* family displays a wide range of sensitivities to abiotic stress and hormones. Several *AsaPP2C* were involved in the plant responses to both saline and ABA stresses to varying degrees, indicating that these *AsaPP2Cs* may have regulatory functions in tolerance to saline and ABA stress and deserve further investigation. Our results provide a foundation for further researching *PP2C* gene functions in *A. sativa*. Future researchers should establish a CRISPR-based genome editing technique to fast-track the development of stress-proof oat cultivars, merging genome-wide PP2C characterization with precision gene-editing blueprints for abiotic stress resilience.

## Figures and Tables

**Figure 1 plants-14-02062-f001:**
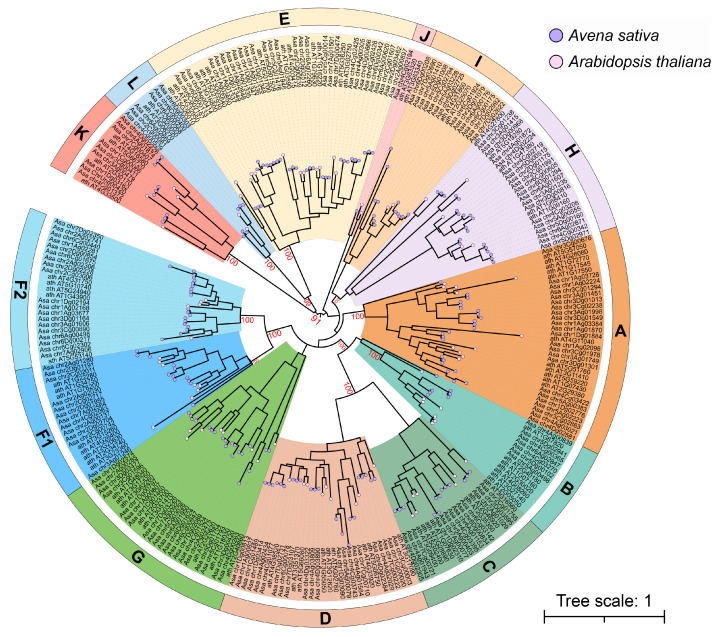
Phylogenetic tree of PP2C family members of *A. sativa* and *A. thaliana* using the maximum likelihood method. The prefix Ath was used before the names of the *A. thaliana* PP2Cs, and the prefix Asa was used before the names of the *A. sativa* PP2Cs. The subfamilies are represented by different colors.

**Figure 2 plants-14-02062-f002:**
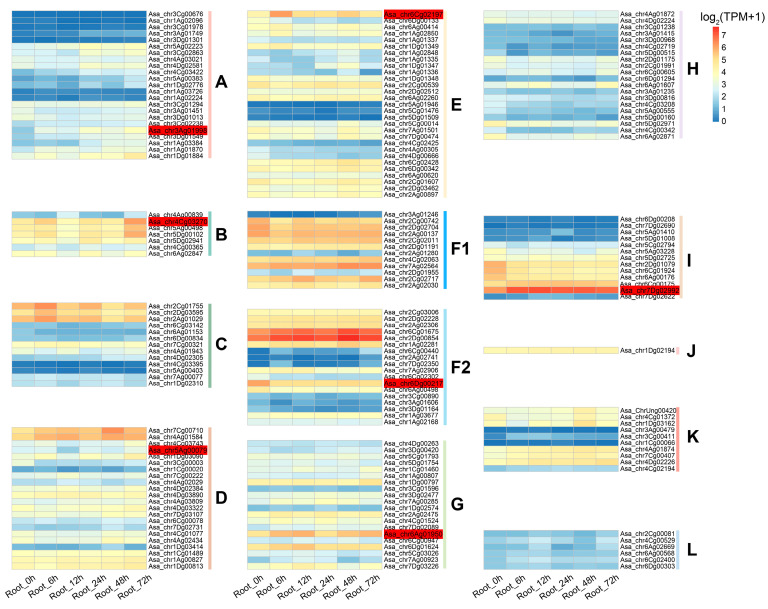
Expression pattern of *PP2C* genes in *A. sativa* under drought stress as determined from RNA-Seq data. The expression level was represented based on the color: red, higher expression; blue, lower expression. *AsaPP2C*, red-labeled, was specifically mentioned in the manuscript.

**Figure 3 plants-14-02062-f003:**
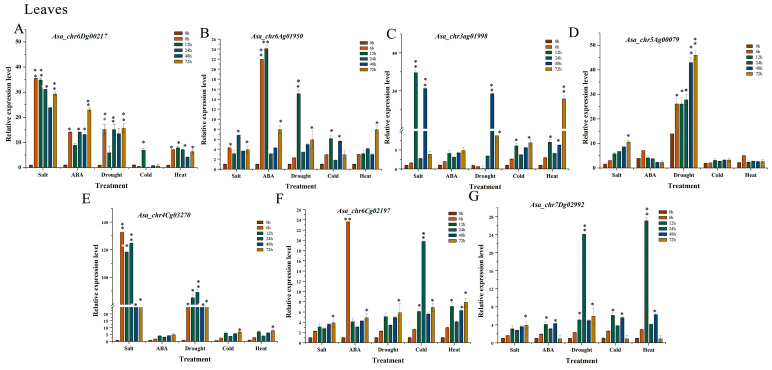
The expression pattern analysis of *AsaPP2C* genes in leaf organs under salt, ABA, drought, cold, and heat stress treatment for 0, 6, 12, 24, 48, and 72 h using qRT-PCR. (**A**) *Asa_chr6Dg00217;* (**B**) *Asa_chr6Ag01950*; (**C**) *Asa_chr3ag01998*; (**D**) *Asa_chr5Ag00079;* (**E**) *Asa_chr4Cg03270*; (**F**) *Asa_chr6Cg02197*; (**G**) *Asa_chr7Dg02992.* Asterisks (**) indicate highly significant differences from control (0h) for each treatment time point, *p* < 0.01, and asterisks (*) indicate significant differences from control (0h) for each treatment time point, *p* < 0.05.

**Figure 4 plants-14-02062-f004:**
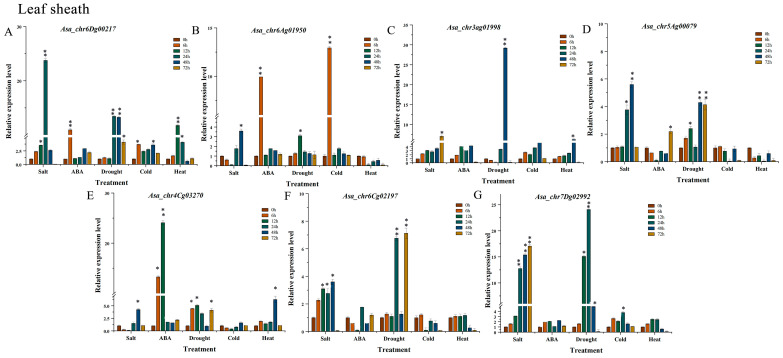
The expression pattern analysis of *AsaPP2C* genes in leaf sheath organs under salt, ABA, drought, cold, and heat stress treatment for 0, 6, 12, 24, 48, and 72 h using qRT-PCR. All data are expressed as the mean ± standard error (SE) of three independent replicates. (**A**) *Asa_chr6Dg00217;* (**B**) *Asa_chr6Ag01950*; (**C**) *Asa_chr3ag01998;* (**D**) *Asa_chr5Ag00079;* (**E**) *Asa_chr4Cg03270*; (**F**) *Asa_chr6Cg02197*; (**G**) *Asa_chr7Dg02992.* Duncan’s analytical test (*p* < 0.01) was used to determine the significance of the differences between 0 h and other points in time. Asterisks (**) indicate highly significant differences from control (0h) for each treatment time point, *p* < 0.01, and asterisks (*) indicate significant differences from control (0h) for each treatment time point, *p* < 0.05.

**Figure 5 plants-14-02062-f005:**
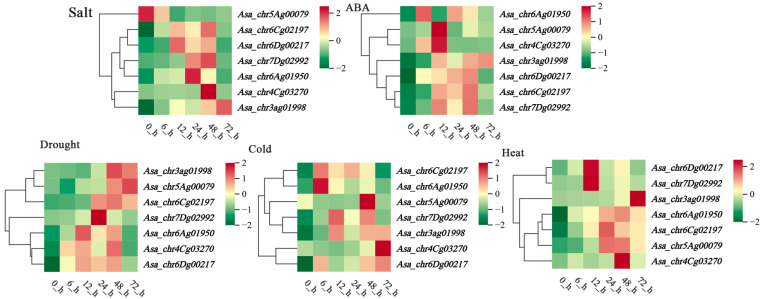
The expression pattern analysis of *AsaPP2C* genes in root organs under salt, ABA, drought, cold, and heat stress treatment for 0, 6, 12, 24, 48, and 72 h using qRT-PCR. The expression level is represented by the color: red, higher expression; green, lower expression.

**Figure 6 plants-14-02062-f006:**
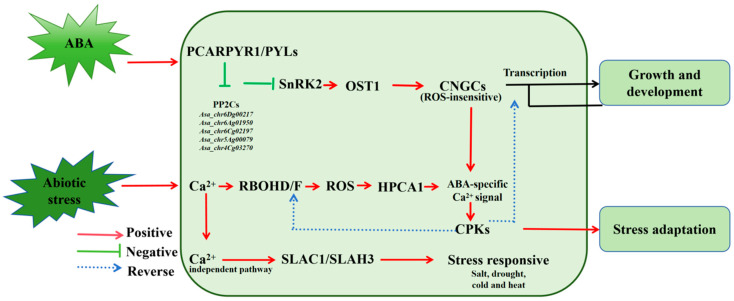
A working model of the response of *AsaPP2Cs* to ABA and abiotic stresses. Under abiotic stress, salt stress leads to a sudden increase in the intracellular Ca^2+^ content, and intracellular calcium signaling, as well as ROS signaling, is activated to increase the expression of genes such as *Asa_chr6Dg00217*, *Asa_chr7Dg02992*, *Asa_chr5Ag00079*, and *Asa_chr4Cg03270* by binding to their promoters to mediate the *A. sativa* response to abiotic stress. And hormone-responsive genes such as *Asa_chr6Dg00217*, *Asa_chr6Ag01950*, *Asa_chr6Cg02197*, *Asa_*chr5Ag0007,9 and *Asa_chr4Cg03270* may act in the regulation of *A. sativa* growth and development through transcription factors that bind to related cis-acting elements in the promoter regions of the genes.

## Data Availability

Data will be made available on request.

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
