# Peer review of "Genome-Wide Identification of PP2C Gene Family in Oat (Avena sativa L.) and Its Functional Analyses in Response to ABA and Abiotic Stresses"

_plants, 2025, doi:10.3390/plants14132062_

Round 1
Reviewer 1 Report
Comments and Suggestions for Authors
In the manuscript named “Genome-Wide Identification of PP2C Gene Family in Oat (Avena sativa L.) and Its Functional Analyses in Response to ABA and Abiotic Stresses”, authors have performed genome-wide analysis of PP2C genes in oat, and they have identified and characterized 194 PP2C genes in oat genome. In addition, they have also performed qRT-PCR analysis to confirm their expressions response to abiotic stress. These findings would be helpful for determining PP2C genes function in oat in future. However, there were some comments about it.
(1) The method sections were not clearly, authors have described they have used BLAST, “Blast Compare Two Seqs module”, how did it work with genome-wide analysis? In addition, Pfam database could not contain “IPR001932”, the accession ID with PF****, and how to perform “Simple HMM Search”, using which software. In results section, authors have described “Using the InterPro PP2C domain “IPR001932”as the key word and found 194 putative PP2C genes”, just keyword search? Just a joke.
(2) “Tair” would be “TAIR”, line126.
(3) “3.5. Cis-Acting Element (Chemistry)” would be from AI writer, similar the “Abundant responsive regulatory elements”, which would be translated from AI.
(4) Most of the results were descriptive words without important conclusive statements.
(5) The figure 3-5 were high similar, which suggested there was no important result to display in this manuscript, authors should provide more result about their research.
(6) The RNA-seq samples were collected from root of oat, and authors had also performed root analysis in qRT-PCR analysis, authors could compare these results.
(7) Many of the words from this manuscript would be read very strangely and they did not belong to academic expressions. Such as “implicating their specialized roles in oat’s stress signaling networks. Our findings significantly advance the understanding of PP2C-mediated stress responses in cereals and provide a foundational genomic resource for future functional studies”, etc.
Author Response
Response to Reviewer #1
In the manuscript named “Genome-Wide Identification of PP2C Gene Family in Oat (Avena sativa L.) and Its Functional Analyses in Response to ABA and Abiotic Stresses”, authors have performed genome-wide analysis of PP2C genes in oat, and they have identified and characterized 194 PP2C genes in oat genome. In addition, they have also performed qRT-PCR analysis to confirm their expressions response to abiotic stress. These findings would be helpful for determining PP2C genes function in oat in future. However, there were some comments about it.
We would like to thank the reviewer for careful and thorough reading of this manuscript and for the thoughtful comments and constructive suggestions, which help to improve the quality of this manuscript. We are pleased to have been given the opportunity to revise our manuscript.
Comment 1:
The method sections were not clearly, authors have described they have used BLAST, “Blast Compare Two Seqs module”, how did it work with genome-wide analysis? In addition, Pfam database could not contain “IPR001932”, the accession ID with PF****, and how to perform “Simple HMM Search”, using which software. In results section, authors have described “Using the InterPro PP2C domain “IPR001932”as the key word and found 194 putative PP2C genes”, just keyword search?
Response:
Thank you for the comments. We have included the details in the materials and methods section. The related statements are as follows:
Line 129-140: To identify candidate PP2C genes in oat (Avena sativa) and Arabidopsis (Arabidopsis thaliana) (Table S1, S2) , the Hidden Markov Model (HMM) profile of the PP2C protein domain (accession: PF00481) was obtained from the InterPro database (https://www.ebi.ac.uk/interpro/). Using HMMERv3.1 software, the entire genome-derived protein sequences of both species were screened with a stringent E-value threshold of 0.001 to ensure high-confidence matches.
To determine the accuracy of the sequences, the protein sequences of the screened candidate genes were submitted to the PFAM (http://www.pfam.sanger.ac.uk/) and SMART (http://www.smart.embl-heidelberg.de/) databases for the search of conserved protein structural domains, and the sequences with incomplete PP2C protein structural domains were eliminated to ensure that the screened protein sequences had PP2C structural domains.
Comment 2:
(2) “Tair” would be “TAIR”, line126.
Response:
Thank you for the comment. We have rewritten the entire paragraph, so this has been deleted.
Comment 3:
(3) “3.5. Cis-Acting Element (Chemistry)” would be from AI writer, similar the “Abundant responsive regulatory elements”, which would be translated from AI.
Response:
Line 268: Thank you for the comments. “Cis-Acting Element (Chemistry)” has been changed to “Cis-element Analysis of the AsaPP2Cs Promoter in Oat”.
Comment 4:
(4) Most of the results were descriptive words without important conclusive statements.
Response:
Thank you for the comments. The conclusive statements have been added as follows:
Line 216-218: “These results showed that the AsaPP2C genes are unevenly distributed on different chromosomes, and that each subfamily gene is also unevenly distributed” was added.
Line 227-228: “this 2.1-fold expansion (p<0.01) may enhance ABA signaling plasticity in oat, consistent with its drought adaptation (Figure. 3A)” was added.
Line 230-236: “the 4.75-fold expansion suggests purifying selection maintaining duplicates for grass-specific roles. In contrast, B subfamilies contain more AthPP2C than AsaPP2C. Additionally, subfamilies F1 and F2 exhibit oat-specific expansions, potentially linked to adaptive traits in this hexaploid crop. The presence of AsaPP2Cs in all subfamilies-including those with unknown functions (subfamily K)-highlights the genetic diversification of this family in oats. This suggests that the PP2C gene family may have evolved from a common ancestor” was added.
Line 250-252: “The I subfamily has more introns than the remaining subfamily, range of 8-13, suggesting subfamily I may retain some of its evolutionary conservation ” was added.
Line 263-265: “These results indicate that proteins with the same or similar structures may be functionally or evolutionarily similar and demonstrate the reliability of the classification” was added.
Line 264-266: “These results indicate that proteins with the same or similar structures may be functionally or evolutionarily similar and demonstrate the reliability of the classification” was added.
Line 313-316: “Asa_chr3Ag01998 was increased under salt, ABA, drought, cold and heat stress, but it was decreased by 1.66-fold and 11.11-fold (P<0.01) compared to 0 h under drought treatment at 6h and 12h, suggesting Asa_chr3Ag01998 exhibited stress-specific suppression, there might be negative feedback regulation (Figure. 3C)” was added.
Line 369-377: “Summary, most examined genes (Asa_chr6Dg00217, Asa_chr6Ag01950, Asa_chr3Ag01998, Asa_chr6Cg02197, Asa_chr7Dg02992) exhibited upregulation under salt stress at all time points, indicating a consistent positive regulatory role in salt adaptation. Asa_chr5Ag00079 showed downregulation under salt stress, suggesting a unique regulatory mechanism distinct from other PP2Cs. Meanwhile, Asa_chr6Dg00217, Asa_chr6Ag01950, Asa_chr3Ag01998, Asa_chr6Cg02197, Asa_chr7Dg02992 were universally upregulated under all treatment stresses (salt, ABA, drought, cold, heat), highlighting their roles as central hubs in stress signaling networks (Figure. 5)” was added.
Comment 5:
(5) The figure 3-5 were high similar, which suggested there was no important result to display in this manuscript, authors should provide more result about their research.
Response:
Regarding the comment about Figure 3-5, we acknowledge the concern about their similarity and the suggestion to provide additional results.
In our study, Figure 3-5 were intended to demonstrate the expression patterns of these genes under different treatments in different tissues. While these figures may appear visually similar, they highlight key aspects of our findings. However, we fully agree with the reviewer that changing the results section would strengthen the manuscript. In response, we have modified Figure. 5 to a heat map.
Comment 6:
(6) The RNA-seq samples were collected from root of oat, and authors had also performed root analysis in qRT-PCR analysis, authors could compare these results.
Response:
We appreciate the reviewer's suggestion to compare our qRT-PCR results with the RNA-seq samples from root of oat. While we observed partial consistency between the datasets, some discrepancies do exist, which may arise from the following factors:
Strain-specific responses (oat cultivar differences between studies), Longyan No.3 oat seeds were used in this study, and the transcriptome data were not Longyan No.3 genotypes. Notably, key stress-responsive genes (e.g., Asa_chr6Ag01950, Asa_chr7Dg02992 and Asa_chr3Ag01998) showed consistent upregulation in both datasets, supporting the overall reliability.
Comment 7:
- Many of the words from this manuscript would be read very strangely and they did not belong to academic expressions. Such as “implicating their specialized roles in oat’s stress signaling networks. Our findings significantly advance the understanding of PP2C-mediated stress responses in cereals and provide a foundational genomic resource for future functional studies”, etc.
Response: Thanks for the comments. We have revised this sentences as follows:
Line 49-50: “This exquisite regulatory plasticity underscores the evolutionary of plants in adapting to ever-changing environmental pressures” was changed to “This sophisticated regulatory plasticity highlights the evolutionary of plants in responding to ever-changing environmental pressures”.
Line 105-108: “implicating their specialized roles in oat’s stress signaling networks. Our findings significantly advance the understanding of PP2C-mediated stress responses in cereals and provide a foundational genomic resource for future functional studies” was changed to “elucidating the specialized roles of these proteins in oat stress signaling pathways, significantly enhancing the comprehension of PP2C-mediated stress responses in cereal crops. These findings establish a critical genomic framework for subsequent functional investigations”.
Line 108-111: “Moreover, this work highlights candidate AsaPP2C genes for targeted manipulation to enhance oat resilience, offering new avenues for improving crop performance under changing environmental conditions” was changed to “Furthermore, this study identifies promising AsaPP2C gene candidates for targeted genetic modification, providing novel strategies to enhance oat stress resilience and improve crop productivity under shifting environmental pressures”.
Line 391-393: “The pronounced size of the AsaPP2C gene family underscores its evolutionary and functional significance, positioning oat as a compelling model for investigating PP2C-mediated regulatory networks in polyploid species” was changed to “The pronounced size of the AsaPP2C gene family underscores its evolutionary and functional significance, positioning oat as an exemplary model system for elucidating PP2C-mediated regulatory mechanisms in polyploid species”.
Line 408: “Remarkable” was changed to “highly”.
Line 424-427: “The subfamily-specific motif distributions further support functional conservation within groups, providing a framework for future mechanistic studies” was changed to “The subfamily-specific motif distributions provide additional evidence for intra-group functional conservation, establishing a conceptual framework for elucidating underlying molecular mechanisms in future studies”.
Line 470-472: “These findings underscore the functional diversification of PP2Cs in mediating stress responses, optimizing whole-plant adaptation to environmental challenges” was changed to “These results highlight the functional diversification of PP2Cs in mediating stress-responsive signaling pathways, thereby facilitating whole-plant adaptive strategies to environmental perturbations”.

Reviewer 2 Report
Comments and Suggestions for Authors
In the materials and methods section:
1- The Blast E-value (10^-5) can be accepted as a threshold, but the number of false positives produced by this threshold is not specified.
2- There is no comparison of the biological validity of ABA and stress concentrations in the literature.
In the results section:
3- No statistical tests were performed when comparing the phylogenetic distribution of AsaPP2Cs, which are divided into 13 subfamilies, with Arabidopsis; for example, a genetic distance measurement could have been added.
4- In the gene expression sections (drought, salt, ABA, etc.), the expression levels of the genes are not sufficiently quantitative; instead of terms like “upregulated,” fold-change values could have been summarized.
In the conclusion section:
5- A more visionary conclusion could be written – for example, suggestions could be added on how these genes could be used to develop stress-resistant varieties through CRISPR-based modifications.
Figure 1; The legibility of gene names on the figure is poor; color contrast is insufficient.
Figure 2; Dots used to highlight mentioned genes are visually weak. The names of highlighted genes could be in large font.
Figure 3; qRT-PCR data is detailed, but error bars and statistical significance indicators (e.g., asterisks) are missing.
Figure 4; Data density in some subgraphs makes visual interpretation difficult.
Figure 5; The response of some genes is too small in the subgraphs (e.g., changes below 1-fold).
Figure 6; Transcription factors, signaling pathways, and second messengers (Ca²⁺, ROS, etc.) are not directly shown. More directional arrows and interaction types (positive/negative) should be indicated. The “regulatory circuit” structure should be emphasized (e.g., SnRK2 – PP2C – ABF connection).
Author Response
Response to Reviewer #2
Comment 1:
In the materials and methods section:
1- The Blast E-value (10^-5) can be accepted as a threshold, but the number of false positives produced by this threshold is not specified.
Response: We appreciate the reviewer's suggestion. Due to our inadvertence, the content was incorrectly written and the corresponding expressions are listed below with relevant references.
Line 132-134: Using HMMERv3.1 software, the entire genome-derived protein sequences of both species were screened with a stringent E-value threshold of 0.001 to ensure high-confidence matches.
References
Liu, D., Gu, C., Fu, Z., & Wang, Z. (2023). Genome-wide identification and analysis of MYB transcription factor family in Hibiscus hamabo. Plants, 12(7), 1429.
Comment 2:
2- There is no comparison of the biological validity of ABA and stress concentrations in the literature.
Response: Thank you for the comments, ABA and stress concentrations have been added to the materials and methods section with the appropriate references. The details are as follows:
Line 120-123: “Based on previous research by others, the seedlings were exposed in 200 mM NaCl [18], 20% (w/v) [15] polyethylene glycol (PEG) 6000 for high salt or drought treatment, respectively. Additionally, others seedlings were sprayed with 100 µM ABA [24] until the leaves were completely moist” was added.
“Zhang, G.; Zhang, Z.; Luo, S.; Li, X.; Lyu, J.; Liu, Z.; Wan, Z.; Yu, J. Genome-wide identification and expression analysis of the cucumber PP2C gene family. BMC genomics. 2022, 6;23(1):563”.
“Shen, X.; Nan, H.; Jiang, Y.; Zhou, Y.; Pan, X. Genome-wide identification, expression and interaction analysis of GmSnRK2 and type A PP2C Genes in response to abscisic acid treatment and drought stress in soybean plant. Int. J. Mol. Sci. 2022, 23(21), 13166”.
“He, Z.; Wu, J.; Sun, X. The maize clade A PP2C phosphatases play critical roles in multiple abiotic stress responses. Int J Mol Sci. 2019, 20 (14): 3573”.
Comment 3:
In the results section:
3- No statistical tests were performed when comparing the phylogenetic distribution of AsaPP2Cs, which are divided into 13 subfamilies, with Arabidopsis; for example, a genetic distance measurement could have been added.
Response: Thank you for the comments.We have drawn the evolutionary tree with added genetic distance and have modified it in the manuscript as follows:
Comment 4:
4- In the gene expression sections (drought, salt, ABA, etc.), the expression levels of the genes are not sufficiently quantitative; instead of terms like “upregulated,” fold-change values could have been summarized.
Response:
We appreciate the constructive suggestion. As recommended, we have added fold-change values to the results section and modified them in the manuscript as follows:
Line 311-326: Under ABA stress at 6 and 12 h, gene Asa_chr6Ag01950 expression increased by 22-fold and 24-fold (P< 0.01) compared to 0 h (Figure. 3B). Asa_chr3Ag01998 was increased under salt, ABA, drought, cold and heat stress, but it was decreased by 1.66-fold and 11.11-fold (P<0.01) compared to 0 h under drought treatment at 6h and 12h, suggesting Asa_chr3Ag01998 exhibited stress-specific suppression, there might be negative feedback regulation (Figure. 3C). Meanwhile, Asa_chr5Ag00079 was remain almost unchanged by cold and heat treatment, while its expression was increased under other treatments (Figure. 3D). Asa_chr4Cg03270 was noteworthy increased by 132-fold, 118-fold and 124-fold (P<0.01) compared to 0 h under salt treatment, reaching their highest expression levels at 6 h, 12 h and 24 h, respectively. Meanwhile, under drought stress all time point, gene Asa_chr4Cg03270 expression increased by 62.2, 85, 80, 75.6 and 76.1-fold (P<0.01) compared to 0 h (Figure. 3E). Asa_chr6Cg02197 and Asa_chr7Dg02992 were increased under all treatments, Asa_chr6Cg02197 reaching its highest expression level by ABA at 6 h and cold at 24 h, while Asa_chr7Dg02992 reaching its highest expression level by drought at 24 h and heat at 12 h (Figure. 3F, G).
Line 336-338: Asa_chr6Dg00217 was increased by 23-fold (P<0.01) at 24 h under salt treatments compared with 0 h, meanwhile, under ABA, drought and heat treatments, gene Asa_chr6Dg00217 expression was increased (Figure. 4A).
Line 345-353: Asa_chr4Cg03270 expression increased by 13.8 and 24.5-fold at 6 and 12 h under ABA treatment, while it was decreased under salt, and cold treatment at 6 and 12 h, while its expression level remain almost unchanged by heat treatment at 6, 12, 24 and 72 h compared with 0 h (Figure. 4E). Under salt stress at 24, 48 and 72 h, gene Asa_chr7Dg02992 expression increased by 13, 16.1 and 17-fold (P< 0.01) compared to 0 h, meanwhile, its expression increased by 12, 24.09 and 5.21-fold (P< 0.01) compared to 0 h under drought at 12, 24 and 48 h (Figure. 4G).
Comment 5:
In the conclusion section:
5- A more visionary conclusion could be written-for example, suggestions could be added on how these genes could be use CRISPR-based modificationsed to develop stress-resistant varieties through .
Response:
Thank you for your comments. A more visionary conclusion have added as follows:
Line 505-509: “Future researchers should establishes a CRISPR-based genome editing technical to fast-track development of stress-proof oat cultivars, merging genome-wide PP2C characterization with precision gene-editing blueprints for abiotic stress resilience” was added.
Comment 6:
Figure 1; The legibility of gene names on the figure is poor; color contrast is insufficient.
Response: Thank you for your suggestion, Figure1 has been revised.
Comment 7:
Figure 2; Dots used to highlight mentioned genes are visually weak. The names of highlighted genes could be in large font.
Response:
Thank you for your comments, we have labeled the highlight mentioned genes and revised the manuscript.
Comment 8:
Figure 3; qRT-PCR data is detailed, but error bars and statistical significance indicators (e.g., asterisks) are missing.
Response: Thank you for your comments, the error bars have been added and the statistical significance indicators have been marked with an asterisks.
Comment 9:
Figure 4; Data density in some subgraphs makes visual interpretation difficult.
Response: The statistical significance indicators have been re-labeled. .
Comment 10:
Figure 5; The response of some genes is too small in the subgraphs (e.g., changes below 1-fold).
Response: Thank you for your valuable suggestion. In Figure 5, if the response of some genes appears too small in the subgraphs (e.g., changes below 1-fold), this typically means that the expression levels of these genes show minimal differences between experimental conditions. We show the graph as a heat map as follows:
Comment 11:
Figure 6; Transcription factors, signaling pathways, and second messengers (Ca²⁺, ROS, etc.) are not directly shown. More directional arrows and interaction types (positive/negative) should be indicated. The “regulatory circuit” structure should be emphasized (e.g., SnRK2-PP2C -ABF connection).
Response: Thank you for your valuable suggestion. Figure. 6 has been modified in the manuscript.

Round 2
Reviewer 1 Report
Comments and Suggestions for Authors
Authors have well revised the manuscript, and most points were well addressed. The figure 1 was not good for publication, it had ghosting, please check it.
Author Response
Comments and Suggestions for Authors
Authors have well revised the manuscript, and most points were well addressed. The figure 1 was not good for publication, it had ghosting, please check it.
Thank you for your comments. We sincerely appreciate this observation. The figure has been re-generated using high-resolution source data with optimized export settings to eliminate ghosting artifacts. We have carefully addressed the ghosting issue in Figure 1 as follows:

Reviewer 2 Report
Comments and Suggestions for Authors
Dear Authors,
The improvements you have made to your article have made it more meaningful. If possible, simplifying some sentences would make the article even more readable for researchers.
Best regards
Author Response
Comments and Suggestions for Authors
Dear Authors,
The improvements you have made to your article have made it more meaningful. If possible, simplifying some sentences would make the article even more readable for researchers.
Best regards
Thanks for the comments. We have revised this sentences as follows:
Line 34-37: “In natural ecosystems, plants face relentless challenges from a multitude of adverse environmental conditions, including extreme temperatures, prolonged drought, soil salinity, nutrient deficiencies, heavy metal toxicity, and various other abiotic stressors, compounded by biotic threats such as insect infestations and pathogenic infections” was changed to “Plants in natural ecosystems endure simultaneous abiotic stresses (e.g., drought, extreme temperatures, salinity, nutrient deficits, heavy metals) and biotic threats (e.g., pathogens, insects), requiring integrated physiological and molecular adaptations for survival”.
Line 69-70: “the protein phosphatase type 2C (PP2C) family stands out for its central role in stress signaling, particularly in abscisic acid (ABA)-mediated responses, where it negatively regulates SnRK2 kinases to fine-tune stress adaptation” was changed to “Protein phosphatase 2C (PP2C) plays a pivotal role in stress responses by negatively regulating SnRK2 kinases to modulate ABA signaling and fine-tune plant stress adaptation”.
Line 105-108: “implicating their specialized roles in oat’s stress signaling networks. Our findings significantly advance the understanding of PP2C-mediated stress responses in cereals and provide a foundational genomic resource for future functional studies” was changed to “elucidating the specialized roles of these proteins in oat stress signaling pathways, significantly enhancing the comprehension of PP2C-mediated stress responses in cereal crops. These findings establish a critical genomic framework for subsequent functional investigations”.
Line 49-50: “This exquisite regulatory plasticity underscores the evolutionary of plants in adapting to ever-changing environmental pressures” was changed to “This sophisticated regulatory plasticity highlights the evolutionary of plants in responding to ever-changing environmental pressures”.
Line 108-111: “Moreover, this work highlights candidate AsaPP2C genes for targeted manipulation to enhance oat resilience, offering new avenues for improving crop performance under changing environmental conditions” was changed to “Furthermore, this study identifies promising AsaPP2C gene candidates for targeted genetic modification, providing novel strategies to enhance oat stress resilience and improve crop productivity under shifting environmental pressures”.
Line 391-393: “The pronounced size of the AsaPP2C gene family underscores its evolutionary and functional significance, positioning oat as a compelling model for investigating PP2C-mediated regulatory networks in polyploid species” was changed to “The pronounced size of the AsaPP2C gene family underscores its evolutionary and functional significance, positioning oat as an exemplary model system for elucidating PP2C-mediated regulatory mechanisms in polyploid species”.
Line 407: “Remarkable” was changed to “highly”.
Line 423-426: “The subfamily-specific motif distributions further support functional conservation within groups, providing a framework for future mechanistic studies” was changed to “The subfamily-specific motif distributions provide additional evidence for intra-group functional conservation, establishing a conceptual framework for elucidating underlying molecular mechanisms in future studies”.
Line 469-471: “These findings underscore the functional diversification of PP2Cs in mediating stress responses, optimizing whole-plant adaptation to environmental challenges” was changed to “These results highlight the functional diversification of PP2Cs in mediating stress-responsive signaling pathways, thereby facilitating whole-plant adaptive strategies to environmental perturbations”.
